# Mixed Reality Meets Robotic Systems: A HoloLens2-Enabled Waypoint Navigation Interface

Albus Dongchen Fang[1], Xiaomeng Xu[2], Yanran Lin[2], Alia Gilbert[1], Dimitra Panagou[1,3]

*Abstract*—User interfaces for robotic systems often force users to shift focus between digital interfaces and their physical surroundings, which leads to inefficiencies and potential safety issues. In this paper we present a novel Mixed Reality system, which by seamlessly integrating holographic elements with the physical world, seeks to overcome these limitations. Our proposed system connects the Microsoft HoloLens2 with an overhead camera and a ground rover to enable mixed reality-based control for robotic navigation. The system allows users to set waypoints for an autonomous rover using a holographic interface displayed through the HoloLens2, providing an intuitive and immersive control experience. The interface is engineered with the objective of augmenting user awareness of both the environmental context and system dynamics, and delivers real-time visual feedback. With this proposed design, we address the challenge of enhancing user multitasking and situational awareness in complex environments. To the best of our knowledge, this is the first open-source mixed reality human supervisory control system supporting waypoint multi-robot control through HoloLens2. The system has been tested by many users from our department and demonstrated during educational and outreach activities on campus (e.g., during lab tours). This paper discusses the system's design, implementation, and user experience, and provides insights into future improvements and applications of mixed reality in robotic control systems.

## I. Introduction

The rapid advancement of mixed reality (MR) technologies has opened new possibilities for Human-Robot Interaction (HRI), enabling more intuitive ways to engage with both physical and digital content. One promising application of MR lies in robotic control, where users can leverage MR interfaces to directly command and monitor robotic systems in real-time. MR presents an opportunity to blend physical environments with digital supervision, allowing users to interact seamlessly with remote systems. For example, consider a warehouse worker manipulating boxes while simultaneously overseeing robots transporting other items across the space. Such systems demonstrate the potential for MR to enhance both hands-on tasks and remote robot supervision.

Several studies have explored integrating MR with remote robot supervision. MR has been shown to improve telepresence, providing users with an enhanced sense of being physically present in a remote environment [1]. While MR

is an emerging field, Virtual Reality (VR) has already become a well-established tool for robotic teleoperation. VR enables users to immerse themselves in either virtual representations of remote locations or entirely simulated worlds, as demonstrated in works like Allspaw (2018) and Kalinov (2021) [2][3][4].

In contrast, MR offers a more hybrid approach, blending the real and virtual worlds. For example, Chen et al. (2024) explored multi-robot manipulation using MR by combining 3D spatial mapping with intuitive drag-and-drop features [5]. Other research, such as Ostanin (2019), has focused on MR-based teleoperation interfaces for robot control [6]. Unlike VR, MR allows users to operate within their existing physical environments while interacting with digital elements, which requires more advanced sensing and mapping capabilities.

Although the idea of using VR to control robots has appeared since as early as 1993 [7], popular implementation of mixed reality for human-robot interaction (HRI) has only begun in the last few years [8], which makes it an open area of research with exciting possibilities and challenges. When MR is used in HRI, the human is often considered a supervisory agent [9].

Mixed Reality has been used to control UAVs and heterogeneous agents within the last few years[10], [11], [12]. However, the interaction between the human and the agents has broadly been more focused on teleoperation, where the human is guiding autonomous agents directly on what to do. If we assume that the robots are more independent in their tasks, then the human's supervisory role can be less frequent. While waypoint navigation has been explored for high-level guidance, [13], [14], the implimentations can be difficult to repeat, and do not provide real time video streaming of the remote supervised area. The interface for monitoring a group of robots and assisting their task assignment is still an open area of research which is directly relevant to the current state of technology.

This paper presents a novel integrated system that combines Microsoft's Hololens2, an overhead camera, and a ground rover to allow the users to control the rover by setting waypoints through an original holographic interface displayed on HoloLens2. When designing the system, we aimed to provide a user-friendly and engaging platform for robotic navigation and make it broadly accessible to users of all ages and skill levels. We discuss the development of the proposed system and specifically its technical components and the user interface design. The discussion focuses on opportunities and limitations of the technical design, as well as future applications.

This work was supported in part by a DEI Faculty Grant from the University of Michigan.

*Emails: {albfang, phoebexu, yanran, galia, dpanagou}@umich.edu.

[1]Department of Robotics, University of Michigan, Ann Arbor, MI, USA.

[2]Department of Computer Science and Engineering, University of Michigan, Ann Arbor, MI, USA.

[3]Department of Aerospace Engineering, University of Michigan, Ann Arbor, MI, USA.

## II. METHODOLOGY

### A. Hardware Summary

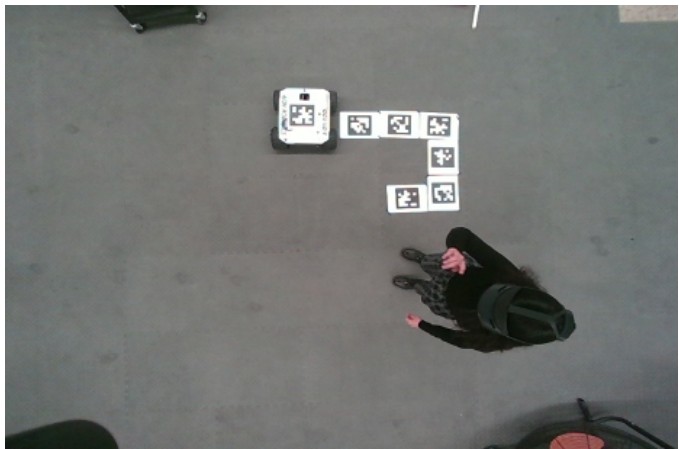

Fig. 1: System picture in which a ground rover, Microsoft HoloLens 2, and Apriltags for camera calibration is captured by an overhead camera

In this section, we provide a detailed discussion of the hardware components that are critical to the system's functionality. As shown in Fig. 2, the system comprises four essential hardware components - an Intel RealSense camera, a Jetson Xavier as the processing unit, a Microsoft HoloLens 2 Mixed Reality device, a ground rover, and one supplemental ground PC for safety measures for the rover.

*1) Global Coordinate System:* We used a Vicon motion capture system to receive the real-time pose of the ground rover. The Vicon system utilizes a number of specialized cameras to track the pose of the rover with 'Vicon pearls' attached and broadcasts the global coordinates of the rover with respect to the Vicon system origin onto a Robot Operating System 2 (ROS2) topic, and we subscribe to this topic in our ROS node detailed later.

*2) Roles of Components:*

1) *Intel RealSense Camera*: This camera captures an aerial view of the ground rover and its immediate surroundings. For future applications, the camera can be attached to a drone to map complex terrains and cover longer distances.
2) *Jetson Xavier*: As the main processing unit, Jetson Xavier performs a variety of tasks such as transforming coordinates, publishing waypoints to ROS topics, and communicating with HoloLens 2.
3) *Microsoft HoloLens 2*: The HoloLens 2 displays a holographic user interface which overlays the live aerial video feed captured by the robot's camera. This interface is designed to facilitate interaction with the ground rover and provides real-time visualization of the robot's traveled path and projected optimal trajectory. Upon activation of the air-tap gesture, the system highlights the current waypoint chosen by the user via air-tapping the HoloLens2 interface that has yet to be reached with a red sphere. Once the waypoint is reached, the sphere's

color changes to green, and the visual marker would be subsequently removed, as shown in Fig. 5.

4) *Ground Robot*: Equipped with Raspberry Pi and other units, the rover receives setpoint data from ROS 2 topics and uses geometric controller to adjust its body velocity and angular velocity.
5) *Ground Computer*: As a precaution, this computer is responsible for arming and disarming the rover.

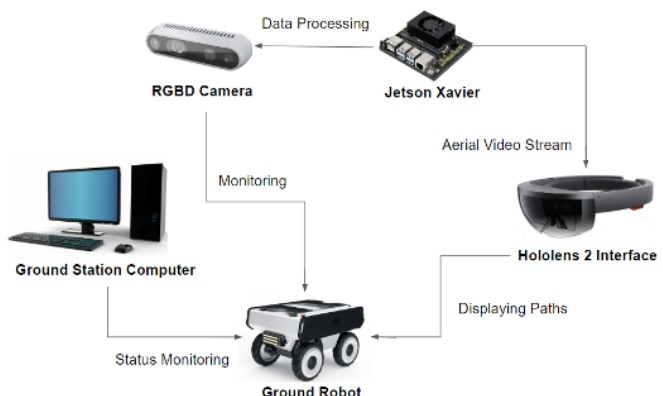

Fig. 2: System Diagram

### B. Camera Calibration

To minimize error induced by inaccuracies when transforming coordinates between the world frame and the camera frame, we calibrate the camera extrinsic matrix with OpenCV and Apriltags. We start by estimating the position and orientation of the camera relative to a known reference frame, in our case the Vicon frame. The process also requires the coordinates of known points, in our case Apriltag corners, in both Vicon frame and pixel frame. OpenCV's calibration functions then use the correspondences between the known 3D points and their 2D projections in the image to compute the extrinsic parameters. These parameters include the rotation and translation vectors, which describe the camera's pose relative to the world frame. Accurate calibration is crucial for enhancing system accuracy and thereby user experience and immersion.

### C. System Algorithm Overview

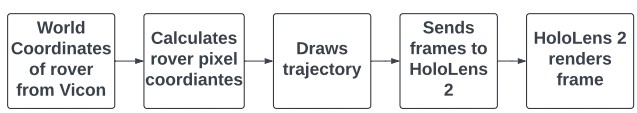

Fig. 3: Static System Flowchart

In this section, we present an algorithm that facilitates communication among three distinct devices: the Jetson Xavier, which functions as the primary processing unit; the HoloLens 2 Mixed Reality device, which provides an interface for the user; and a rover that autonomously operates based on a geometric control policy. The Jetson Xavier runs two concurrent

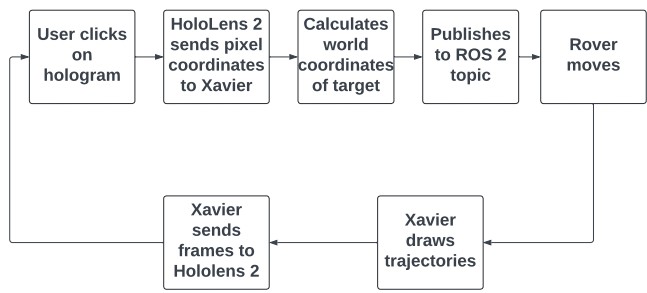

Fig. 4: System Flowchart upon User Click

threads: the first is a Python-based ROS2 Node, while the second monitors messages on a socket channel.

### D. Python ROS2 Node

This ROS2 Node serves as the central algorithmic component for enabling communication between Jetson Xavier, HoloLens 2, and the ground robot. This section provides an overview of its operational details.

*1) Initialization:* To initialize the system, the algorithm sets up the RealSense camera, configures the server socket for communication, and initializes the ROS2 node with necessary parameters. The camera captures both color and depth streams at a resolution of 640 x 480 with a frame rate of minimum 30 FPS. The server socket is established on a specified IP address and port to allow for incoming connections from a client.

*2) Subscription and Publishing:* The ROS2 node subscribes to two topics that provide coordinate transformation data obtained from the Vicon motion capture system: one for the camera's position and orientation and another for the rover. The node also sets up a publisher to send trajectory setpoints to the ground rover. These setpoints are used to guide the ground rover to the desired location based on user inputs. The algorithm employs a timer to invoke the main loop at a frequency of 30 Hz, ensuring timely updates and processing.

*3) Main Loop and Callbacks:* Utilizing a timer function, the algorithm continuously captures video frames from the RealSense camera and updates the world coordinates of the rover in real time based on Vicon data. The algorithm then transforms the rover coordinates received from Vicon to camera frame, and projects this coordinate to the pixel frame. To ensure clarity, a circle whose center is the pixel coordinate of the rover, would be drawn on the frame. The formulae from transforming cooridnates from world frame to pixel frame and vice versa are shown in (1) and (2)

$$\mathbf{P}_{\text{pixel}} = \mathbf{K} \cdot \mathbf{E} \cdot \mathbf{P}_{\text{world}} \tag{1}$$

where $\mathbf{P}_{\text{pixel}}$ and $\mathbf{P}_{\text{world}}$ are the homogeneous coordinates in pixel and world frames respectively.

$$\mathbf{P}_{\text{world}} = \mathbf{E}^{-1} \cdot Z_{\text{camera}} \cdot \mathbf{K}^{-1} \cdot \mathbf{P}_{\text{pixel}} \tag{2}$$

where $Z_{\text{camera}}$ is the depth value of the corresponding pixel coordinate.

*4) Target Detection and Trajectory Planning:* In our proposed system, the process for target tracking and visualization involves several key steps. When a target is set by the user through the HoloLens 2, the algorithm determines if the rover has reached the target by calculating the Euclidean distance between the rover's current pixel coordinate and the target's pixel coordinate. Once the target is reached, the algorithm resets the *target* variable to *None*, while continuously transmitting video frames to the HoloLens 2, awaiting a new target.

Additionally, the system maintains a continuous record of the rover's movement by appending the current position to a list, which is subsequently used to render a dotted trajectory on the color image for visualization. To prevent the trajectory from being cluttered with repeated coordinates when the rover remains stationary, the system limits the number of consecutive identical coordinates that are appended to the trail. Any excess coordinates are discarded to maintain clarity.

To ensure that the length of past trajectories remains manageable, the system periodically removes a number of older coordinates from the trail. However, in the context of demos where participants utilize the trail to draw specific patterns, such as the letter 'M', the system is configured to retain all trajectory coordinates for accurate pattern representation and utilize solid lines when tracing past trajectories.

### E. Video Streaming

We build a pipeline for streaming video images from an Intel RealSense Camera on a Jeson Xavier to HoloLens2 using a Transmission Control Protocol/Internet Protocol (TCP/IP) socket. The camera is configured to capture color frames at a resolution of 640 × 480 pixels at 30 frames per second. Once the image frame from RealSense camera is captured by Jetson Xavier, it will be converted to a NumPy array and then encoded into JPEG format using OpenCV. Once the HoloLens is succesfully connected, Xavier will start transmitting each captured frame over the network by first sending the frame size and then the image data. The streaming continues until disrupted. It is essential that both the HoloLens2 and Jetson Xavier are connected to the same network to establish a socket communication channel. The use of TCP/IP is to ensure the reliable transmission of data, which are well-suited to the requirements of our application.

### F. Secondary Thread for Socket Communication

A secondary thread is employed to manage incoming target coordinates from the client, allowing for dynamic modification of ROS node parameters without disrupting the main processing loop, which consistently sends video frames to HoloLens 2. This thread operates on a dedicated port and thereby ensures that real-time processing in the main loop remains uninterrupted and the system's responsiveness is preserved. When the secondary thread receives a message, it processes the data by converting the message to a tuple representing the pixel coordinates of the user-selected target point. The *target* variable of the ROS node is then updated with these coordinates, which in turn directs the rover to advance toward the new target. This architecture optimizes system efficiency by segregating data handling from real-time operations.

## G. Interface Design Considerations

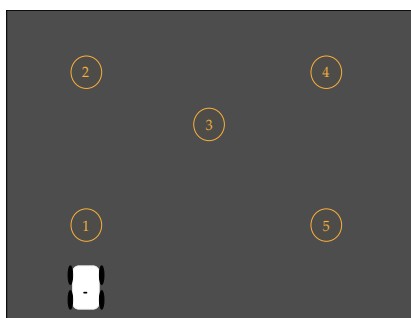

(a) This figure shows the initial user interface on which an aerial view of the experiment area is displayed

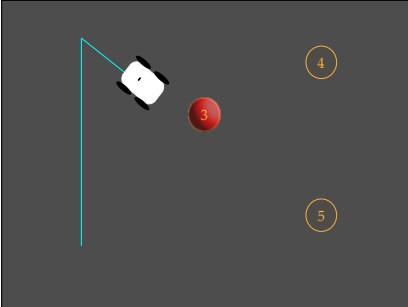

(b) This figure depicts the user interface showing the rover's past trajectory as indicated by the solid blue line. The current waypoint is represented by the red sphere.

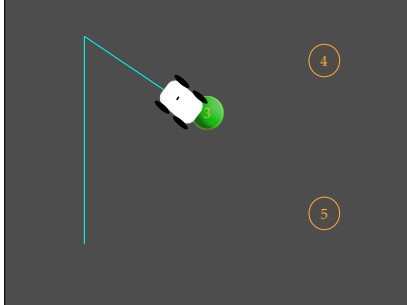

(c) This figure illustrates the user interface display upon the rover's arrival at the designated waypoint. This interface provides real-time feedback by turning the sphere green, indicating the successful attainment of the current waypoint. The sphere and the circled number would disappear afterwards

Fig. 5: The HoloLens 2 interface displays five predefined waypoints. When a waypoint is selected, a red sphere indicates the target position for the rover. Upon the rover reaching the waypoint, the sphere turns green and disappears along with the waypoint marker. The gray area represents the floor mat that the rover operates on.

We focused on the ease of use and real-time visual feedback. For the demo to interested users, we identified five waypoints arranged to trace an 'M'. These numbered waypoints, marked with yellow circles and clearly visible against a black floor mat

background, guide the user by indicating the rover's intended path. When a user selects a waypoint via air-tapping, a red sphere appears to denote the target destination. The rover then travels in a straight line towards this target, following a geometric control policy that aligns with the path of 'M'. Upon reaching the waypoint, the red sphere changes to green and disappears, providing immediate visual feedback that the target has been successfully reached. This visual marker system is designed to keep users informed about the rover's progress and its current destination, thereby enhancing overall interface clarity and user experience. In the proposed algorithm, the system determines the vehicle's proximity to the target by computing the Euclidean distance between the vehicle's current position and the target coordinates. If this distance falls below a predefined threshold, the ROS2 node on Jetson Xavier identifies that the rover has reached its destination. Consequently, it communicates with the HoloLens 2 via a socket connection, instructing the interface to update the visual display by changing the color of the sphere to green and removing the sphere visual marker.

### H. Rover Control

To adapt the system for demos where participants would use the rover's trajectories to follow desired patterns, e.g., draw the letter 'M', we implemented a specialized rover controller designed to accurately trace sharp corners and straight lines. We achieved this by integrating a Rotate-Translate-Rotate (RTR) controller with a geometric controller. The RTR controller governs the rover's initial rotation, ensuring it first turns in place until its heading aligns with the target coordinate, with the angular velocity $\omega$ determined by the difference between the current and target headings. Once the desired heading is achieved, the rover advances in a straight line towards the target under the policy of the geometric controller, which dynamically adjusts the heading and trajectory to maintain straight-line motion.

### I. Safety Measures

We have incorporated several measures for the safety of the users to make the system accessible to various age groups and experience levels. The first feature is a verification mechanism for user-selected destinations based on proximity thresholds around predefined waypoints. The system computes the Euclidean distance between the current unreached waypoint and the user's click coordinate in terms of pixels. If this distance is below a predefined threshold, the system automatically designates the center of the current predefined waypoint as the destination. Conversely, if the distance exceeds the threshold, the user is prompted to reattempt the selection via an air-tap gesture. This approach ensures accurate destination selection while minimizing user errors.

To ensure accurate tracing of the letter 'M' in our system demos, an additional feature prevents the selection of a new waypoint until the current one has been reached. This is achieved by blocking new waypoint selections while the rover is en route. The implementation utilizes a loop within the socket communication thread, which suspends the reception

of new messages until confirmation that the current waypoint has been reached. This mechanism ensures that the rover completes its journey to each waypoint in sequence, thereby maintaining the integrity of the path and improving overall navigation reliability.

## III. Discussion

### A. System Integration and Observations

Our system establishes a wireless connection between the HoloLens2 and the Jetson Xavier via WiFi, utilizing TCP socket communication. Through this connection, the pixel coordinates corresponding to the user's input on the HoloLens2 are transmitted to the Jetson Xavier and video frames captured by the camera are transmitted from Xavier to HoloLens2 continuously. The Vicon system broadcasts real-time position of the ground rover through a ROS2 topic, which the Jetson Xavier subscribes to. After receiving the user's selected coordinates, the Jetson Xavier computes the corresponding target world coordinates for the rover, which are then published on a dedicated ROS2 topic, allowing the rover to navigate towards the specified location. For outdoor applications, the system can be adapted by mounting the Jetson Xavier and a downward-facing camera onto a drone, providing aerial visual feedback and enabling more flexible operations in unstructured environments.

Some key design choices were made. We chose TCP over User Datagram Protocol (UDP) when establishing a channel between HoloLens2 and Jetson Xavier. UDP features lower overhead at the cost of reliability and a connectionless protocol. TCP, on the other hand, is a connection-oriented protocol, which ensures both sides are ready to exchange data before commencing. TCP ensures that all data is delivered to the destination in the correct order, and if a packet is lost or corrupted during transmission, TCP will detect this and retransmit the packet. It also provides error-checking mechanisms. Due to the nature of our application where user engagement is essential, the reliability that TCP features is more desirable. Secondly, we opted for real-time video stream over point cloud mapping for several reasons. The applications of our system demands a low-latency visual feedback. Video stream requires lower processing power than point clouds and induces less latency, whereas point cloud mapping requires more bandwidth due to the high density of 3D points, which makes real-time transmission more challenging. Furthermore, the camera in our setup is considerably more affordable than sensors required for point cloud mapping, such as LiDAR. Owing to these features, our systems can be easily and affordably replicated and deployed which was the objective of this project.

Another advantage of our system is that unlike traditional systems that often confine users to a single mode of interaction (e.g., through a computer screen or a mobile device), Mixed Reality systems offer a more holistic approach. The HoloLens 2 is capable of projecting holographic interfaces into the user's physical environment. This integration of the interface with the physical world enables a simultaneous awareness of with both virtual and real-world elements. The primary benefit of this dual interaction is the enhanced multitasking capability it

affords users. Traditional systems typically necessitate users to switch focus between their digital tasks and their physical environment. This often leads to inefficiencies and potential safety hazards, particularly in dynamic or complex environments where situational awareness is crucial, such as disaster rescue scenarios. In contrast, Mixed Reality systems like the HoloLens 2 allow users to maintain a continuous visual connection to their surroundings while interacting with digital overlays. This feature ensures that users can manage tasks without sacrificing awareness of their immediate environment.

The integration of HoloLens2 with the overhead camera and the ground rover, and particularly the synchronization between the holographic interface and the rover's movement, ensure that the interactive experience of the user is smooth and responsive. These features are critical in maintaining user engagement and trust in the system's capabilities. Some improvements for future research would include further reducing the latency caused by transmitting frames from Jetson Xavier to HoloLens 2, enhancing the responsiveness of the system as we do observe the existence of mild latency.

Our current system setup necessitates that the HoloLens 2 and Jetson Xavier devices be connected to the same network to establish socket connection, a requirement easily met in an indoor setting by utilizing a shared Wi-Fi connection. However, further investigation may be warranted to ensure robust communication between the devices in various networking environments. In our controlled indoor laboratory setting, the high-speed network has facilitated timely transmission of messages and video frames between the devices. Nonetheless, additional research and testing are encouraged to minimize latency and ensure optimal performance in outdoor environments, where network conditions may vary.

### B. Applications

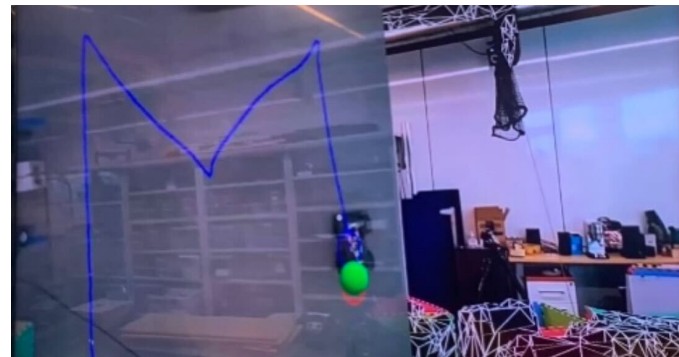

Fig. 6: This pictures displays a successful execution of a system demo in which the final waypoint has just been reached. As shown in the picture, the interface allows the user to visualize the holographic interface and the physical surroundings.

*1) System Demonstrations and Educational Potential:* To engage and inspire potential users to such systems, we have been actively demonstrating the system to our local community in the department and during lab tours and other outreach events on campus. We have also invited interested users to

interact with the built-in demos. Interested users participate in a thirty-minute session during which they are first introduced to a Hololens 2 tutorial, which aims to familiarize them with the air-tap gesture crucial for interacting with the interface. The air-tap gesture, a fundamental feature of the HoloLens 2 system, involves pointing to a location on the interface with the index finger and then performing a pinching motion with the thumb and index finger to make a selection. This gesture is used to set waypoints within the interface. After completing the basic tutorial, the user is given the opportunity to control a ground rover and navigate to the waypoint shown in Figure 5a. From our (admittedly limited, so far) experience, we can note that the enthusiasm of the users is immediately evident upon their introduction to the system. Many express a strong desire to learn more about the system, with several people actively seeking the opportunity to participate in the demo. The intuitive gesture-based controls and real-time feedback provided by the system makes the experience both enjoyable and educational. We have often received follow-up questions and comments that the participants were highly appreciative of the system and expressed a strong interest in learning more about its underlying components. Specifically, the holographic interface has been commented to be user-friendly and accessible: the participants feel that it is intuitive to navigate the holographic interface. This was attributed in part due to the design of the Hololens 2 interface combined with the clear visual representation of the rover's surroundings and waypoints, which allowed participants to intuitively understand and control the rover's movements. Furthermore, visual markers, such as the sphere representing the rover's objective and traveled trajectories, provided constant situational awareness and were instrumental in ensuring participants remained oriented throughout the task. While the feedback and comments are just at the level of testing the system during lab experiments and demos, they may still be a promising indication of the potential that such technologies in educational activities in an effective and inclusive manner, since such technologies need to cater to diverse age groups and varying levels of prior knowledge.

*2) Natural Disaster Response:* Natural Disaster Response is another domain where our system can provide significant assistance. In a disaster stricken area, our system can be deployed to assist rescuers to locate victims in hazardous situations such as collapsed buildings or unstable terrains, and help the victims survive while awaiting the rescuers. The user can direct a ground rover carrying essential items such as food or water to the victim via the holographic interface. The drones, fitted with cameras, autonomously explore the disaster area, capturing and streaming real-time aerial footage directly into the user's HoloLens2 workspace. The video stream allows the rescue operators to maintain continuous visibility of the region, enabling them to make informed, split-second decisions. Additionally, owing to the features discussed before, the system tracks the ground rover's movements and projects a dotted line on the holographic interface to represent its past trajectory. Simultaneously, a solid line illustrates the calculated optimal path toward the identified target location. This real-time feedback allows rescuers to monitor the rover's

progress and adjust the mission as conditions evolve.

In such scenarios characterized by dynamic and unpredictable environments, situational awareness could be crucial in ensuring operators' safety. Traditional user interfaces involve a computer-based design that require users' constant physical presence. This setup forces users to frequently shift focus between their displays and the surrounding environment, which can be both cumbersome and detrimental to situational awareness. Our system has the potential to overcome or minimize such limitations owing to the nature of MR devices. The first advantage is that MR devices are light-weight wearable technologies that are extremely portable. As opposed to a laptop or a desktop, which are bulky and impractical for highly mobile and demanding environments, MR devices offer a hands-free experience that allow the users to adjust their location with ease, enabling a dynamic workspace. This mobility allows disaster response teams to remain agile and adaptive, crucial for operations that require quick reactions and real-time decision-making. MR devices also allow our system to project a digital holographic interface directly on the 3D world. This capability not only streamlines task execution but also enhances safety by keeping users informed of environmental changes or hazards without needing to divert their attention away from their surroundings.

*C. Future Work and Other Applications*

The potential applications of this system are extensive and impactful across various domains, such as educational and outreach programs to the broader community, natural disaster response, and agricultural management. To maximize the system's effectiveness in these scenarios, further enhancements could include integrating advanced path planning and mapping algorithms. The overhead camera, integrated with a drone platform, can autonomously explore the designated area, employing computer vision algorithms to detect and identify victims or crops in real-time. In an outdoor environment, the local coordinate tracking Vicon system we currently employ would be substituted with GPS, which may introduce increased positional inaccuracies. To mitigate this, sensor fusion techniques can be utilized, integrating GPS data with accelerometer readings to enhance state estimation accuracy. These improvements would enable more sophisticated navigation and operational capabilities, making the system more versatile. The promising potential of this technology underscores the need for continued development and refinement through research advances and extensive user studies.

## IV. Conclusion

Through leveraging the novel technology of MR devices, we designed and implemented a system that integrates a Microsoft Hololens2, a ground robot, and an overhead camera, in order to enable the direct command of ground robots via a Hololens2 interface. Based on the initial (admittedly limited) interaction and feedback during demos of the system, we outlined the potential impact of this project and its future applications across various domains, including educational and outreach activities, natural disaster response and agricultural

management. Additionally, we indicated some potential steps that can assist researchers to achieve these objectives towards further research and development in the overlapping areas of MR devices and robotics.

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
