# OpenReview forum: "Mixed Reality Meets Robotic Systems: A Hololens2-Enabled Waypoint Navigation Interface"
_humanrobotinteraction.org/HRI/2025/Workshop/VAM — HRI 2025 Workshop VAM Submission_

### Official Review · Reviewer_8Hwk · 2025-02-28

**Rating:** 7
**Confidence:** 5

**Review:**

Very interesting paper that explores MR interfaces with robotic systems. This submission holds high relevance to the workshop contents.
However, here are a couple of suggestions to improve the paper and its readability:
In the related works section, often terms are introduced without proper explanation or reference. Consider further elaborating on terms such as ''heterogenous agents''
At times, other works/authors were referenced without a clear indication of what is referenced or what their respective study was about. Consider expanding on why these works are relevant to your study
Some visual readability can be augmented such as the numbering list of ''roles of component'' is confusing

---

### Decision · Program_Chairs · 2025-02-26

Accept